# Mutational patterns in chemotherapy resistant muscle-invasive bladder cancer

David Liu [1,2], Philip Abbosh[3], Daniel Keliher[1,2], Brendan Reardon[1,2], Diana Miao[1,2], Kent Mouw[1], Amaro Weiner-Taylor[2], Stephanie Wankowicz[1,2], Garam Han[1,2], Min Yuen Teo[4], Catharine Cipolla[4], Jaegil Kim[2], Gopa Iyer [4], Hikmat Al-Ahmadie[4], Essel Dulaimi[3], David Y.T. Chen[3], R. Katherine Alpaugh[3], Jean Hoffman-Censits[5], Levi A. Garraway[1,2], Gad Getz [2], Scott L. Carter[1,2], Joaquim Bellmunt[1,2], Elizabeth R. Plimack [3], Jonathan E. Rosenberg[4] & Eliezer M. Van Allen [1,2]

Despite continued widespread use, the genomic effects of cisplatin-based chemotherapy and implications for subsequent treatment are incompletely characterized. Here, we analyze whole exome sequencing of matched pre- and post-neoadjuvant cisplatin-based chemotherapy primary bladder tumor samples from 30 muscle-invasive bladder cancer patients. We observe no overall increase in tumor mutational burden post-chemotherapy, though a significant proportion of subclonal mutations are unique to the matched pre- or post-treatment tumor, suggesting chemotherapy-induced and/or spatial heterogeneity. We subsequently identify and validate a novel mutational signature in post-treatment tumors consistent with known characteristics of cisplatin damage and repair. We find that post-treatment tumor heterogeneity predicts worse overall survival, and further observe alterations in cell-cycle and immune checkpoint regulation genes in post-treatment tumors. These results provide insight into the clinical and genomic dynamics of tumor evolution with cisplatin-based chemotherapy, suggest mechanisms of clinical resistance, and inform development of clinically relevant biomarkers and trials of combination therapies.

---

[1] Dana-Farber Cancer Institute, Boston, MA 02215, USA. [2] Broad Institute of Harvard and MIT, Cambridge, MA 02142, USA. [3] Fox Chase Cancer Center, Philadelphia, PA 19111, USA. [4] Memorial Sloan Kettering Cancer Center, New York, NY 10065, USA. [5] Thomas Jefferson University Hospital, Philadelphia, PA 19107, USA. Joaquim Bellmunt, Elizabeth R. Plimack, Jonathan E. Rosenberg and Eliezer Van Allen contributed equally to this work. Correspondence and requests for materials should be addressed to E.R.P. (email: elizabeth.plimack@fccc.edu) or to J.E.R. (email: rosenbj1@mskcc.org) or to E.M.V.A. (email: eliezerm_vanallen@dfci.harvard.edu)

Cisplatin-based chemotherapy remains a mainstay of treatment in many cancers[1], yet the genomic effects of cisplatin-based chemotherapy and implications for subsequent treatment strategies, including immune checkpoint blockade[2], are incompletely characterized. Since a standard of care for muscle invasive bladder cancer (MIBC) is neoadjuvant cisplatin-based chemotherapy followed by cystectomy[3–5], pretreatment tumor biopsy and post-chemotherapy cystectomy specimens are clinically available, creating an ideal setting to identify genomic effects specifically of cisplatin-based treatment in MIBC. Furthermore, it has been proposed that a combination of chemotherapy and immune therapy may result in synergistic efficacy[2] through multiple mechanisms, including increased mutation and neoantigen load induced by DNA-damaging agents such as cisplatin. In this setting, understanding the effect of chemotherapeutic agents in clinical samples is critical to inform the design of combination or sequential therapies. A recent study of genomic heterogeneity in matched primary muscle invasive bladder cancer (MIBC) and mixed post-treatment metastatic and primary tumors have revealed complex phylogenies and clonal selection in advanced urothelial cancers[6]. We hypothesized that a genomic assessment of patient-matched pre- and post-platinum based chemotherapy specimens from the same anatomic site may allow deeper characterization of features specifically associated with treatment exposure itself (rather than features associated with the metastatic process), and that these features may directly inform the role of cytotoxic chemotherapy in this setting.

Towards that end, we examined a cohort of MIBC patients who underwent cisplatin-based neoadjuvant chemotherapy followed by cystectomy and analyzed genomic tumor changes in matched pre- and post-treatment bladder tumor samples.

Our primary results include the finding that mutational and neoantigen load is not significantly increased after therapy in matched post-treatment primary tumor samples despite DNA-damaging therapy. We observe mutations private to pre- and post-treatment samples and find that they are primarily subclonal, suggesting chemotherapy-induced selection though sampling heterogeneity likely also plays a part. To further dissect chemotherapy-induced mutations, we perform a mutational signature analysis and discover a novel signature in post-cisplatin-based-chemotherapy tumor samples sharing features with a pre-clinical experimentally derived cisplatin-induced mutational signature, and consistent with known characteristics of cisplatin damage and repair. We then validate this signature in an additional independent cohort of pre- and post-cisplatin therapy matched urothelial carcinomas. We further find that increased intratumoral heterogeneity is an independent negative predictor of overall survival in this chemotherapy-resistant cohort adjusting for clinical covariates. Finally, we observe alterations in cell-cycle and immune checkpoint regulation genes in post-treatment tumors. These results provide insight into the clinical and genomic dynamics of tumor evolution with cisplatin-based chemotherapy, suggest mechanisms of clinical resistance, and inform

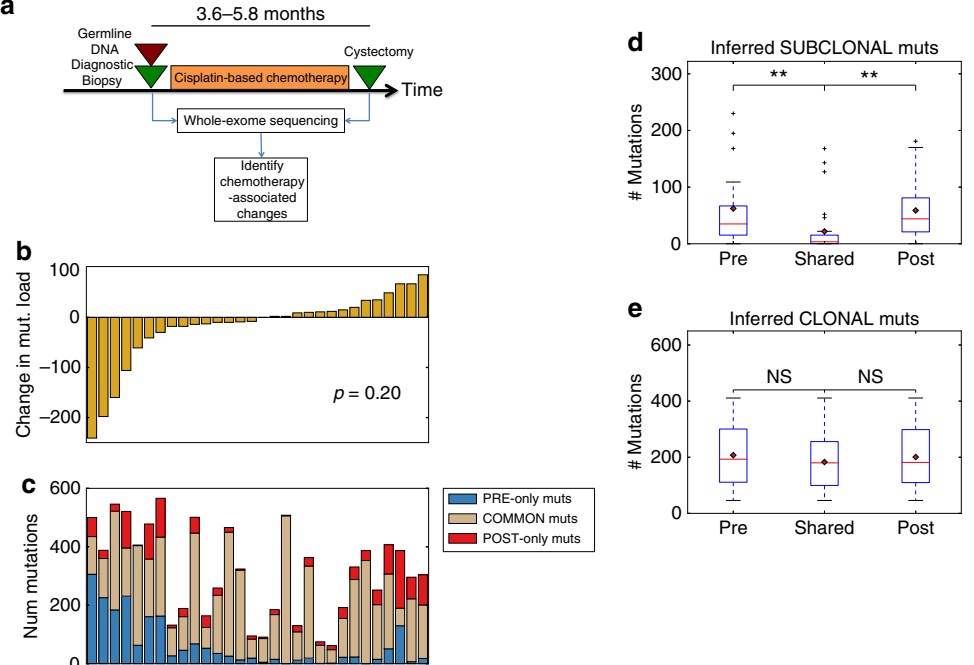

**Fig. 1** Overall chemotherapy associated changes in genomic alterations. **a** Schematic overview of tumor tissue collection in the context of neoadjuvant cisplatin-based chemotherapy, followed by whole-exome sequencing and analysis. The 25th to 75th percentile time between diagnosis and cystectomy samples was 3.6–5.8 months. **b** Inferred changes in mutational load per patient from pre-treatment to post-treatment. Overall, there is no statistically significant change in the total mutational load (mean change = −17.3, paired *t*-test *p* = 0.20). **c** Breakdown of mutations private to the pre-treatment tumor, post-treatment tumor, and common to both. The mean number of "pre-only" mutations (private to pre-treatment tumor) and "post-only" mutations (private to post-treatment tumor) mutations is 64.7 (SD = 81.1) and 47.5 (SD = 46.9), respectively. **d** Boxplot of pre-treatment, shared, and post-treatment subclonal mutations. There are almost no shared subclonal mutations (median 3.5 mutations, 25th–75th percentile 1–15 mutations). There is a statistically significant difference between the number of inferred subclonal pre-treatment and shared, and shared and post-treatment mutations (Mann–Whitney U *p* = 1.2e-04, *p* = 1.9e-05 respectively). **e** Boxplot of pre-treatment, shared, and post-treatment clonal mutations. There is no statistically significant difference between the number of inferred clonal mutations in the pre-treatment tumors and shared between pre and post-treatment tumors, and shared mutations and post-treatment mutations (Mann–Whitney U *p* = 0.38, *p* = 0.51). SD Standard Deviation, N.S. Not statistically significant; "*"Indicates *p* < 0.05; "**"Indicates *p* < 0.01

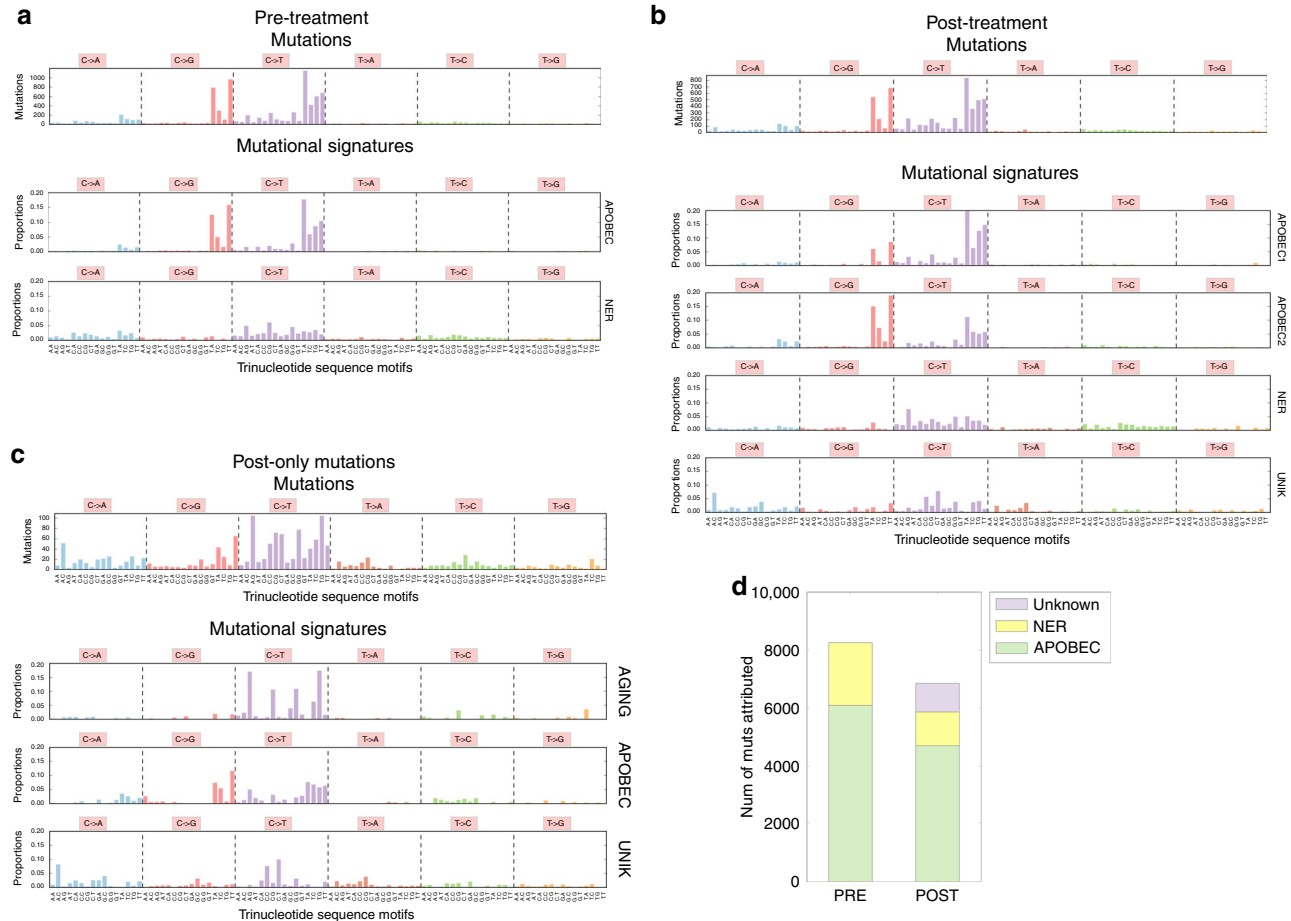

**Fig. 2** Mutations and mutational signatures in pre- and post-treatment tumors. **a** Mutations and mutational signatures in pre-treatment tumors. The top bar graph shows the number of mutations per trinucleotide sequence motif across all pre-treatment tumors. The bottom two bar graph shows the mutational signatures inferred in pre-treatment tumors. Two mutational signatures, matching previously described mutational signatures associated with APOBEC activity (cos sim = 0.99 with average of COSMIC Sig 2 and 13, = 0.82 with each individually), and nucleotide excision repair (NER) deficit (cos sim = 0.91 with COSMIC 5), were discovered. The NER signature also appears to have the aging signature embedded at a low level, which our data is unable to resolve. **a** Mutations and mutational signatures in post-treatment tumors. The top bar graph shows the number of mutations per trinucleotide sequence motif across all post-treatment tumors. The bottom bar graphs show the mutational signatures inferred in post-treatment tumors. Four mutational signatures were discovered, with three matching previously described signatures (APOBEC1: cos sim = 0.96 with COSMIC Sig 2; APOBEC2: cos sim = 0.95 with COSMIC Sig 13; NER: cos sim = 0.86 with COSMIC Sig 5) and an additional signature (UNK) not matching any previously described signature. **b** Mutations and mutational signatures from post-only mutations. Here we consider only those post-treatment mutations not found in matched pre-treatment tumors. The top bar graph shows the number of mutations per trinucleotide sequence motif, and the bottom graphs show mutational signatures inferred in these mutations. Three signatures were inferred, two of which matched previously discovered signatures (AGING: cos sim = 0.86 with COSMIC Sig 1; APOBEC: cos sim = 0.90 with average of COSMIC Sig 2 and 13; UNK: an additional unmatched signature). **c** Inferred mutational signature activity in pre-treatment tumors and post-treatment tumors. The majority of mutations are inferred to be due to mutational signatures associated with APOBEC activity and nucleotide excision repair (NER) deficit, but 14% of mutations in the post-treatment tumors are associated with an unknown mutational signature

## Results

**Patient cohorts and genomic landscape.** From two prospective clinical trials and a previously published cohort[7–9], we identified a cohort of MIBC patients who underwent neoadjuvant cisplatin-based chemotherapy (gemcitabine and cisplatin (GC); or methotrexate, vinblastine, adriamycin, and cisplatin (MVAC)) and subsequent cystectomy. We performed whole exome sequencing on matched pre-chemotherapy biopsy tissue, post-chemotherapy cystectomy tumor tissue, and peripheral blood as a germline control (Fig. 1a) in patients with resistant disease at cystectomy (n = 56). After quality control (Methods), results from

development of clinically relevant biomarkers and trials of combination therapies.

30 non-responders were available for analysis using multiple analytical pipelines (Methods). Relevant demographic, treatment, and tumor characteristics are summarized in Supplementary Data 1 and 2. In our cohort, the most frequently altered driver genes[10] were *TP53* (68%), *KMT2D* (28%), *CDKN2A* (23%), *ARID1A* (22%), *PIK3CA* (22%), and *RB1* (20%) (Supplementary Fig. 1).

**Changes in mutational and neoantigen load after chemotherapy.** We hypothesized that DNA-damaging chemotherapy may lead to increased mutational load in post-treatment tumor. However, we observed no statistically significant change in mutation load in our cohort (mean change = −17.3 mutations, paired *t*-test p = 0.20, Fig. 1b). Examination of specific mutations

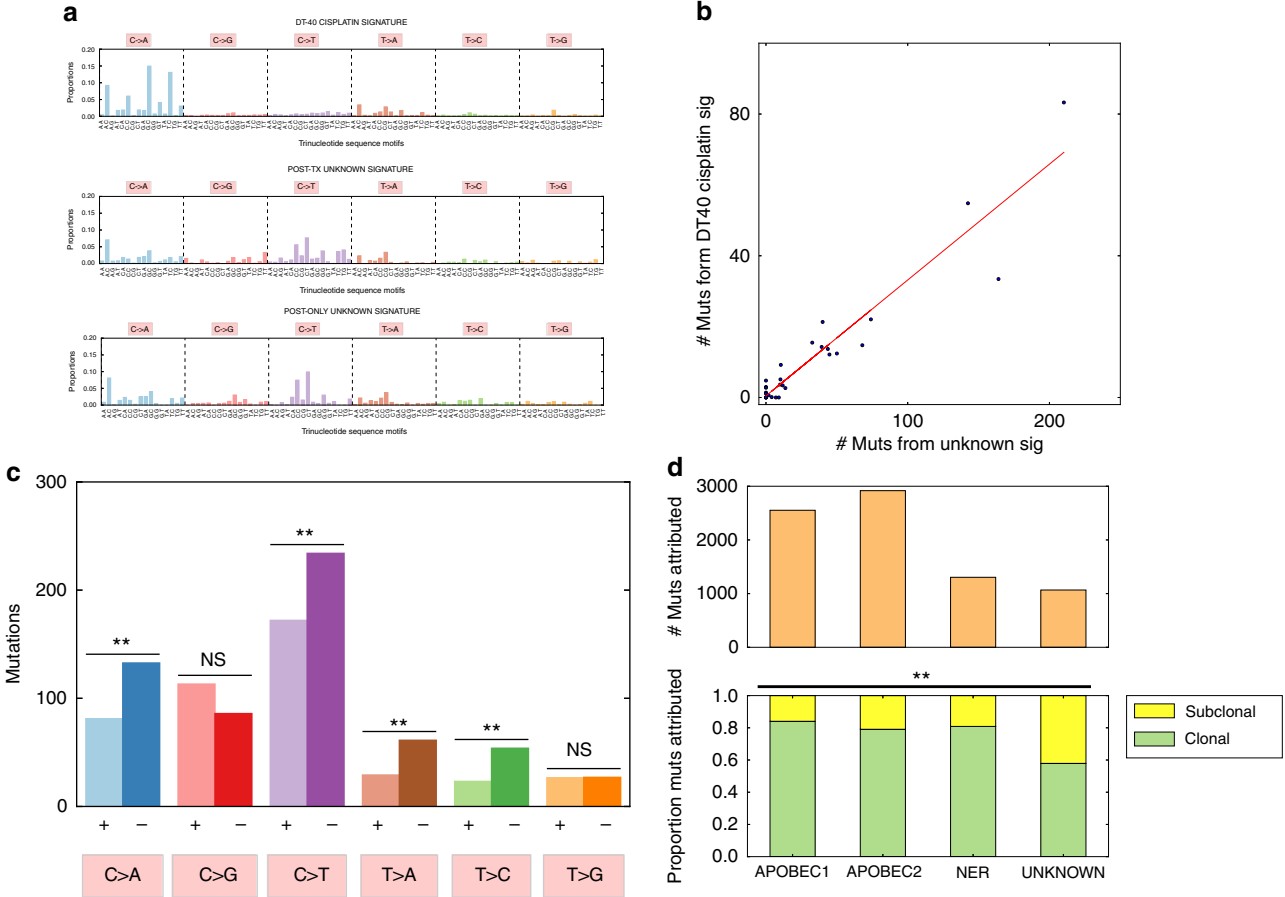

**Fig. 3** Evaluation of a potential cisplatin chemotherapy mutational signature. **a** Comparison of a chicken lymphoblast (DT40) cisplatin-induced mutational signature and the two unknown signatures found in the post-treatment and post-treatment-only mutational signatures. These are mutations induced in chicken lymphoblast (DT40) cells treated with cisplatin[18], adjusted for relative trinucleotide motif frequencies between the chicken and human genomes/exomes. The cos similarity of this signature with the candidate cisplatin signature is modest (0.58), but there are similar T > A motifs (cos sim = 0.87), and the two unknown signatures are similar to each other (cos sim = 0.90). **b** Correlation between DT40 cisplatin signature activity and the unknown signature activity in the same post-treatment tumors. We replaced the unknown signature with the DT40 cisplatin signature, and inferred its activity (Methods), yielding a Pearson correlation coefficient of 0.948 ($p = 0.004$ and $p = 0.049$ for a null distribution of correlations generated by replacing our candidate signature with permutations of the DT40 signature and combinations of COSMIC signatures, respectively (Methods)). **c** Transcriptional strand bias in the unknown signature. There was evidence of transcription strand bias in C > A ($p = 5.8e-04$), C > T ($p = 0.003$), T > A ($p = 0.001$), and T > C ($p = 7.6e-04$) (binomial test with null probability = 0.5). A bias against coding (+) strand mutations in C > X and T > X mutations is consistent with transcription coupled repair of platinum crosslinking in the non-coding strand at GpG and ApG motifs, which together represent 90% of crosslinking sites[21,22]. **d** Association of the unknown signature with subclonal mutations. We inferred relative proportions of clonal and subclonal mutations in post-treatment tumors attributed to each mutational signature in post-treatment tumors. The unknown signature has greater proportion of subclonal mutations (42%) compared to the overall average (22%) (chi-squared $p = 6.2e-68$ with DF = 3). N.S. = Not statistically significant; "*"Indicates $p < 0.05$; "**"Indicates $p < 0.01$

indicated that there were a significant proportion of mutations private to (i.e., observed only in) pre- or post-treatment tumors; the average number of mutations private to pre-treatment and post-treatment tumors was 64.7 (standard deviation (SD) of 81.1) and 47.5 (SD 46.9), respectively (Fig. 1c). Characterization of mutations as clonal or subclonal (Methods) revealed that the majority of private mutations were subclonal (Fig. 1d), and there were very few shared subclonal mutations (median = 3.5 mutations, 25th–75th percentile 1–15 mutations; $p = 1.2e-04$ and $p = 1.9e-05$ (Mann–Whitney U) for the difference between pre-treatment and shared, and shared and post-treatment subclonal mutations, respectively). On the other hand, no statistically significance difference in the number of clonal pre-treatment, shared, and post-treatment mutations was observed ($p = 0.38$, $p = 0.51$ (Mann–Whitney U), Fig. 1e). While this finding may be consistent with chemotherapy-induced subclone reduction and new mutation generation, the confounding effects of tumor

spatial heterogeneity (as observed in multi-regional biopsies[6,11,12]) cannot be ruled out with this analysis alone. A similar analysis of change in predicted neoantigens (Methods) from pre- to post-treatment tumors showed high concordance with the change in mutational load (Pearson rho = 0.85, Supplementary Fig. 2a), and a trend towards decreased number of neoantigens (mean change = −41.8 neoantigens, $p = 0.07$ (paired $t$-test with true mean of 0), Supplementary Fig. 2b).

**Changes in global copy number alterations after chemotherapy**. We next examined global copy number alterations (CNAs) in our cohort (Methods), and found that the CNA landscape in matched pre- and post-treatment tumors was very similar (Supplementary Fig. 3) in the majority of tumors. In a few tumors (e.g., FCCC-017), there are significant differences between pre- and post-treatment CNA profiles, which may reflect

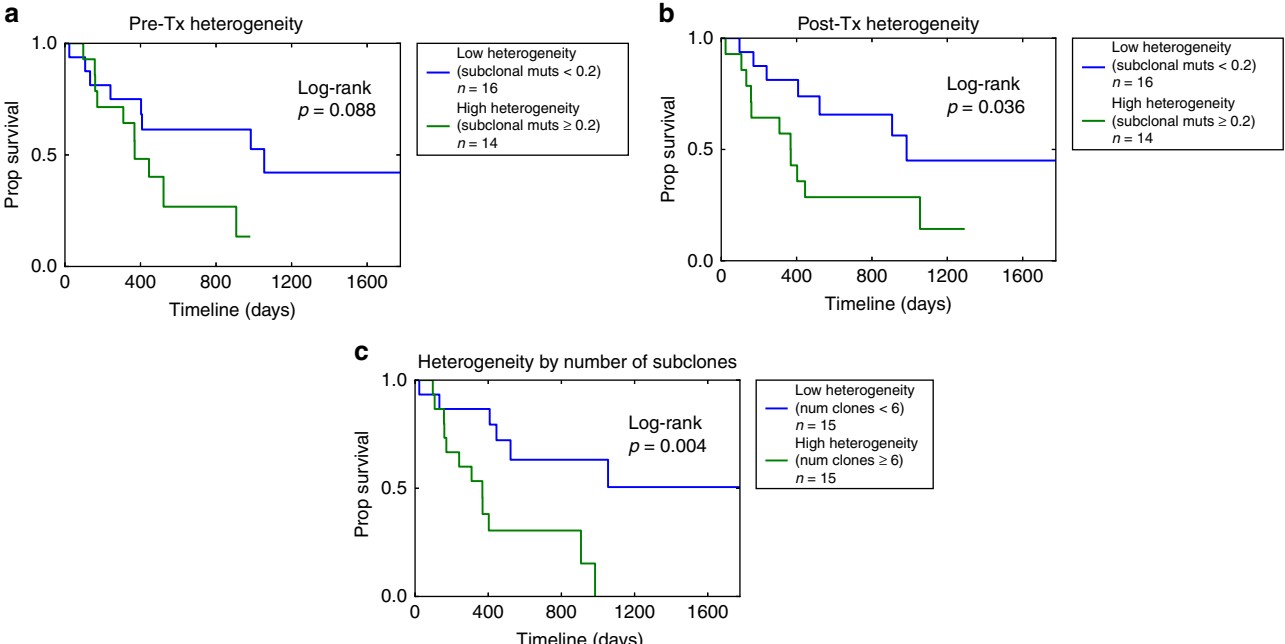

**Fig. 4** Intratumoral heterogeneity association with overall survival. **a** Overall survival of high and low pre-treatment intratumoral heterogeneity. We defined heterogeneity as the proportion of mutations per pre-treatment tumor that were inferred to be subclonal. In Cox proportional hazards analysis, pre-treatment heterogeneity was statistically significantly associated with overall survival (Cox PH: HRR 1.50 (95% CI 1.01–2.23), $p = 0.046$). Dividing the cohort above and below a cutoff of 20% of subclonal mutations ($n = 16/14$ low/high heterogeneity tumors, Supplementary Fig. 12a) resulted in a trend towards improved survival for pts with low pre-treatment heterogeneity (log-rank $p = 0.088$). **b** Overall survival of high and low post-treatment intratumoral heterogeneity. We defined heterogeneity as the proportion of mutations per post-treatment tumor that were inferred to be subclonal. In Cox proportional hazards analysis, post-treatment heterogeneity was negatively associated with overall survival (HRR 1.89 (95% CI 1.1–3.1), $p = 0.013$), and dividing the cohort above and below a cutoff of 20% of subclonal mutations ($n = 16/14$ low/high heterogeneity tumors, Supplementary Fig. 12b) resulted in improved survival for pts those with low post-treatment heterogeneity (log-rank $p = 0.04$). **c** Overall survival of high and low heterogeneity tumors. We defined heterogeneity as the number of inferred subclones (Methods), which includes shared subclones and subclones private to either the pre-treatment or post-treatment tumor. The number of subclones was negatively associated with overall survival (Cox PH: HRR 1.64 (95% CI 1.08–2.49), $p = 0.02$), and using a threshold of 6 clones ($n = 15/15$ with high/low heterogeneity as defined, Supplementary Fig. 12c) demonstrates improved survival for patients with low heterogeneity (log rank $p = 0.004$)

differences in coverage and tumor purity which affect our ability to detect CNAs, but also suggest early evolutionary divergence between the pre- and post-treatment sampled tumors. However, on average, the proportion of the exome with CNAs in the matched pre- and post-treatment tumor only differed by 0.4% ($p = 0.71$ (paired $t$-test with true mean of 0)) with an inter-quartile range of 5% (25th percentile −2.4%, 75th percentile + 2.6% difference, Supplementary Fig. 4).

**Mutational signature discovery.** To further examine chemotherapy-specific effects on post-treatment tumors, we hypothesized that exposure to cisplatin-based chemotherapy may promote mutagenesis and drive a specific pattern of mutations unique to post-exposure tumors. We inferred signatures of mutational activity using non-negative matrix factorization (NMF) on single nucleotide mutations characterized in a tri-nucleotide context[13–15] (Methods). We separately analyzed pre-treatment tumors (Fig. 2a), post-treatment tumors (Fig. 2b), and mutations unique to post-treatment tumors (Fig. 2c), hypothesizing that this last group would be enriched for cisplatin-chemotherapy induced mutations. We identified previously implicated MIBC mutational signatures[6,13,16,17] associated with APOBEC activity and nucleotide excision repair (NER) pathway defects (Fig. 2a, b). In post-chemotherapy tumors, we also identified an additional novel mutational signature (Fig. 2b), which was recapitulated in the analysis of private post-treatment mutations (Fig. 2c) (cos sim = 0.90 between the two unknown

signatures). Overall, 14% of post-treatment mutations were associated with this unknown mutational signature (Fig. 2d).

**Comparison to experimental cisplatin induced mutational signature.** To test the hypothesis that this unknown signature could be a human cisplatin-induced mutational signature, we compared this signature to a mutational signature of cisplatin activity experimentally derived from a chicken lymphoblast (DT40) cell line[18] (DT40) (Fig. 3a, Methods). The human unknown and the DT40 cisplatin mutational signature had modest cosine similarity (0.58), but both were enriched for activity in T > A and C > A contexts compared to the other bladder cancer mutational signatures (Fig. 2b, c). To further examine whether the experimentally derived DT40 mutational signature and our clinically observed unknown signature were functionally related, we examined the correlation between the mutational activity of the unknown signature and inferred mutational activity substituting the DT40 signature for the unknown signature (Methods) (Supplementary Fig. 5) and found a strong correlation (Pearson's rho = 0.949) (Fig. 3b). To determine the likelihood of seeing a correlation of this magnitude or greater by chance, we generated a null distribution of correlations using two approaches: by replacing our unknown signature with permutations of the DT40 signature ($n = 10,000$); and linear combinations of previously discovered mutational signatures[16] ($n = 10,000$) (Methods). We found that the correlation between the DT40 cisplatin signature and our unknown signature was

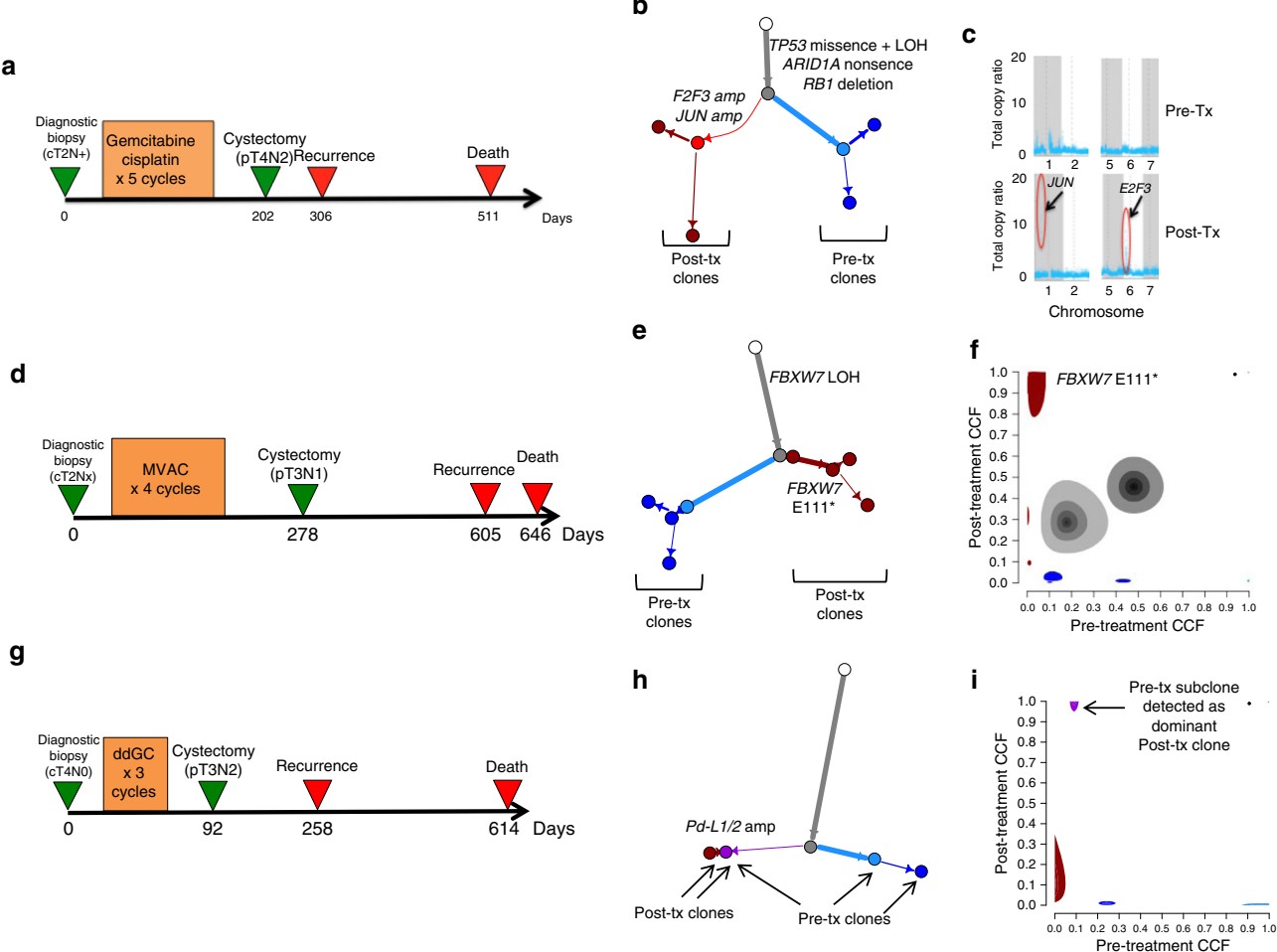

**Fig. 5** Pre and post-chemotherapy changes in individual cases. Patient 1—MSK-0745. **a** Clinical course of Patient 1 with no response to therapy, rapid recurrence and death. **b** Inferred phylogenetic tree showing acquired amplification of *E2F3* and *JUN* in the post-treatment clones. **c** Graph of total genomic copy number showing focal copy number amplifications in segments containing *JUN and E2F3* in the resistant tumor. Patient 2—DFCI-12. **d** Clinical course with no response to chemotherapy and recurrence and death at 605 and 646 days after initial biopsy, respectively. **e** Phylogenetic analysis demonstrating *FBXW7* LOH in the pre-treatment tumor with subsequent additional *FBXW7* E111* event detected in the post-treatment tumor. **f** Pre- and post- treatment cancer cell fractions (CCFs) of inferred tumor subclones. The subclone containing the *FBXW7* E111* mutation had post-treatment CCF of 0.94 but was not detected in the pre-treatment tumor (CCF 0). The size of each subclone cloud represents the uncertainty in the inferred subclone's pre- and post-treatment CCFs. Patient 3—FCCC-022. **g** Clinical course demonstrating poor response to neoadjuvant chemotherapy, recurrence at 258 days and death at 614 days. **h** Phylogenetic analysis demonstrating a gain in *PD-L1/2* in the resistant sample, and that the resistant clone arose out of a pre-existing clone. **i** Pre- and post- treatment CCFs of inferred tumor subclones, with pre-treatment CCF on the x-axis and post-treatment CCF on the y-axis. A subclone with CCF 0.09 in the pre-treatment sample had CCF 1.0 in the post-treatment sample. The size of each subclone cloud represents the uncertainty in the inferred subclone's pre- and post-treatment CCFs

statistically significant (empiric *p*-value = 0.004 and 0.049, respectively) (Supplementary Fig. 6a, b).

**Transcription strand bias and subclonality of mutational signature.** Because cisplatin crosslinks are known substrates for transcription coupled repair[19,20], with crosslinks on the non-coding (−) strand detected and repaired during transcription, we looked for evidence of transcriptional strand bias in the unknown signature. As 90% of cisplatin crosslinks occur at GpG and ApG motifs[21,22], we expect a transcriptional strand bias[14] with decreased C > X and T > X mutations on the coding (+) strand. We repeated the NMF signature discovery process considering mutations on coding and non-coding strands separately (Supplementary Fig. 7), and found evidence of depletion of coding (+) strand mutations in C > A (*p* = 5.8e-04), C > T (*p* = 0.003), T > A (*p* = 0.001), and T > C (*p* = 7.6e-04) (binomial test *p*-values for probability of 0.5) contexts (Fig. 3c).

Next, we hypothesized that cisplatin-induced mutations in our cohort are more likely to be subclonal if associated with temporal emergence in exposed tumors, and found that our unknown signature was indeed associated with a higher proportion of subclonal mutations (42% vs. overall 22%, chi-squared *p* = 6.2e-68) (Fig. 3d).

**Mutational signature validation in an independent patient cohort.** Finally, we performed a mutational signature discovery analysis in an independent validation cohort of 15 urothelial carcinomas with matched pre- and post- cisplatin-chemotherapy samples[6], a total of 19 post-treatment tumors representing a mix of cystectomy (*n* = 10) and metastatic (*n* = 9) samples. In this validation analysis, we focused on mutations unique to the post-treatment tumor, and found a mutational signature with similar motifs and overall concordance to our unknown mutational signature (cos sim = 0.86, Supplementary Fig. 8). Taken together,

these findings suggest that this unknown mutational signature may represent a cisplatin-induced mutational signature in MIBC and indicate direct effect of chemotherapy on mutagenesis.

**Loss and acquisition of specific gene alterations through chemotherapy.** We next looked for loss and acquisition of alterations from pre- to post-treatment tumors in significantly mutated[23] genes but found no alterations that passed multiple hypothesis test correction (Supplementary Fig. 9a, b). We also compared the proportion of tumor samples in the pre- and post-treatment setting with alterations in genes involved in homologous recombination (HR) (Supplementary Fig. 10) and nucleotide excision repair (NER) (Supplementary Fig. 11), hypothesizing that subclones with defects in these pathways may be selected against in the setting of platinum chemotherapy. However, in both HR (Pre-tx: 8/30 altered tumors, Post-tx: 8/30 altered tumors with alterations) and NER (Pre-tx:8/30 altered tumors; Post-tx: 7/30 altered tumors), we did not see a significant difference.

**Intratumoral heterogeneity and survival.** Given the global mutagenesis patterns described above, we then hypothesized that overall intratumoral heterogeneity itself may be associated with survival[24–26] in our resistant cohort absent specific highly recurrent genomic mediators of resistance. We defined intratumoral heterogeneity as the proportion of mutations that were subclonal in a tumor sample (Methods), and considered pre- and post-treatment tumor heterogeneity separately (Fig. 4a, b, Supplementary Fig. 12a, b). In our cohort of resistant disease, high post-treatment heterogeneity was associated with decreased overall survival (Cox PH: HRR 1.89 (95% CI 1.1–3.1), $p = 0.013$) (Fig. 4b), with each 10% increase in proportion of subclonal mutations associated with a 6.6% increase in mortality rate. Pre-treatment heterogeneity was also statistically significantly associated with overall survival in the Cox PH analysis (HRR 1.50 (95% CI 1.01–2.23), $p = 0.046$), though the Kaplan Meier log-rank test was only borderline significant ($p = 0.088$, Fig. 4a). Interestingly, there was only a modest correlation between pre- and post-treatment tumor heterogeneity (Pearson rho = 0.29, $p = 0.12$, Supplementary Fig. 12d). Further, when we adjusted for clinical covariates (age, gender, type of chemotherapy, pathologic staging), post-treatment tumor heterogeneity but not pre-treatment tumor heterogeneity continued to be statistically significantly associated with overall survival (Cox PH: HRR 1.77 (95% CI 1.06–2.97), $p = 0.03$ and HRR 1.38 (95% CI 0.81–2.35), $p = 0.24$ for post- and pre-treatment heterogeneity, respectively, Supplementary Table 1a and b). We then considered the total number of subclones (private and shared) (Supplementary Fig. 12c, Methods) in the pre- and post-treatment tumor as a measure of tumor heterogeneity integrating both pre- and post-treatment tumors, and found that it was significantly associated with overall survival (Cox PH: HRR 1.64 (95% CI 1.08–2.49), $p = 0.02$) (Fig. 4c, Supplementary Fig. 6c), interpreted as an estimated 64% relative increase in rate of mortality for each additional inferred subclone, an association which persisted when adjusted for clinical covariates (HRR 1.80 (95% CI 1.13–2.88), $p = 0.014$; Supplementary Table 1c). Taken together, this suggests that intratumoral heterogeneity predicts worse survival in patients with resistant disease in MIBC.

**Specific genomic alterations in chemotherapy resistant tumors.** Although we did not detect highly recurrent gene alterations predicting resistance to cisplatin-based chemotherapy, we performed phylogenetic analyses of our pre- and post-treatment samples and examined individual cases for insights regarding

molecular subtypes that emerge through cisplatin-based chemotherapy (Supplementary Fig. 13). For example, in one patient (Fig. 5a–c) treated with neoadjuvant gemcitabine and cisplatin with no response to treatment, rapid recurrence and short survival, we identified amplification in focal segments containing *E2F3* and *JUN* (drivers of cell cycle progression) exclusively in the post-treatment tumor sample. Another patient (Fig. 5d) with no response to chemotherapy and early progression and death had a pre-treatment single copy deletion of *FBXW7*, with an additional *FBXW7* E111* mutation event detected in the post-treatment tumor (Fig. 5e, f), The acquisition of a bi-allelic loss of *FBXW7*, a tumor suppressor gene which regulates[27] protein degradation of multiple onco-proteins including c-MYC, Notch, Cyclin E, and c-JUN, in the resistant tumor suggests that this event may play a role in resistance.

Finally, in a third patient with poor response and early death (Fig. 5g), we observed a focal amplification of *PD-L1/2* from pre- to post-treatment tumor (Fig. 5h, Supplementary Fig. 14). In this tumor, a pre-treatment subclone developed into the primary clone in the post-treatment tumor (Fig. 5i, Methods), suggesting that a *PD-L1/2* amplified subclone may have been selected under treatment.

## Discussion

Broadly, clonal evolution has been studied in multiple contexts[6,11,28–30]. This study conducts a focused assessment of mutational changes at a primary bladder tumor site exposed to a standard chemotherapy regimen over a limited time period, providing an opportunity to isolate cisplatin-chemotherapy associated genomic changes in this clinical context. Tumor mutational load has been associated with response to immune checkpoint blockade[31–35], and it has been hypothesized that combining mutation-inducing platinum chemotherapy with immunotherapy would improve response and outcomes[36,37]. However, our data does not support this specific rationale for combination therapy, as we do not find an increase in overall mutational or neoantigen load with cisplatin-based chemotherapy, which is also consistent with a prior study[6] which examined pre- and post-treatment mutational load in a mixed cohort of pre- and post-treatment primary and metastatic urothelial tumors.

We do find evidence of a cisplatin chemotherapy-induced mutational signature. There have been a number of past studies inferring a cisplatin-induced mutational signature. One of the first studies[38] to examine the genomic impact of chemotherapy performed whole genome sequencing on *C. elegans* exposed to a variety of chemotherapies including cisplatin. Interestingly, though this also showed an accumulation of C > A mutations similar to DT40 chicken lymphoblast cisplatin-induced mutations[18], the trinucleotide context is quite different (Supplementary Fig. 15) favoring CpCpC and CpCpT contexts, with a poor overall similarity (cos sim = 0.44). Past studies in human clinical samples[25,39–41] have also identified enrichment in C > A mutations in post-platinum treated samples, though there was study heterogeneity in cancer types (esophageal adenocarcinoma, head and neck squamous carcinoma), platinum therapies (oxaliplatin and cisplatin), and accompanying agents (epirubicin, capecitabine, and 5-FU). In this study, we have inferred a candidate cisplatin-induced mutational signature from a cohort of patients with time-limited pre- and post- cisplatin-based treatment serial biopsies, and carefully derived additional characteristics of this signature including transcription strand bias, increased subclonality, and additional prominent motifs including C > T mutations particularly in CpCpC and CpCpT contexts and T > A mutations in a CpT context. Further, we were able to reproduce this signature in an independent patient cohort[6].

While our mutational signature analysis indicated the presence of a chemotherapy-associated mutational signature in MIBC, further validation through experiments evaluating the effect of cisplatin exposure in human cell lines and matched pre- and post-cisplatin therapy across human tumors would be informative. Further, patients in our cohort received two different cisplatin-containing regimens, which may have different mutation-inducing profiles. Larger clinical cohorts of patients treated with different cisplatin-containing regimens (e.g., MVAC vs. GC) such as SWOG 1314 may help tease apart these differences, as well as mutational studies of combination chemotherapies in human cell lines. If validated, the generation of neoantigens by cisplatin-based chemotherapy may have implications for subsequent immunotherapies or combination regimens, though overall mutational load given these treatment regimens in this clinical context was unchanged. Further, we observe heterogeneity in the inferred level of candidate cisplatin signature activity, suggesting heterogeneity of resistance mechanisms in these tumors. Some tumors may resist platinum treatment through upregulation of efflux pumps, DNA damage repair, or anti-apoptotic signaling. If validated, a cisplatin-associated mutational signature may be useful to differentiate mechanisms of resistance in post-treatment tumors.

We also observed acquired or selected genetic alterations in genes associated with cell cycle checkpoints and regulators (E2F3, JUN, FBXW7), suggesting potential resistance as well as a pro-liferation advantage in the setting of cisplatin chemotherapy that warrant functional evaluation. FBXW7 also regulates MYC, which has been functionally demonstrated to promote immune evasion through direct upregulation of PD-L1 and CD47[42]. Altogether with findings suggesting selection of a PD-L1/2 amplified sub-clone in the setting of cisplatin chemotherapy, this lends support to the hypothesis that chemotherapy may exert anti-tumor effects through modulation of the immune system[43]. Further studies examining the tumor immunological landscape and response to chemotherapy through histologic or transcriptomic (e.g., single-cell) analyses in pre- and post-treatment settings may clarify this relationship further. Further, PD-L1/2 amplification and expression has been observed to be a biomarker of sensitivity to immune checkpoint blockade[44,45]. Although rare in urothelial cancers, it may be predictive when present.

Finally, consistent with prior results[6], we find significant intratumoral heterogeneity in our cohort, and further find that heterogeneity (particularly post-treatment) predicts overall survival in our chemotherapy-resistant cohort. Prior work[24–26,46] found that intratumoral heterogeneity (variably defined) was associated with overall survival and response to immune check-point therapy. While this may reflect intrinsic tumor biology, our results suggest that post-treatment intratumoral heterogeneity (perhaps reflecting broad tolerance to treatment) may provide additional prognostic information; both pre- and post-treatment assessments of intratumoral heterogeneity may aid in risk stra-tification and should be explicitly assessed in future clinical trials.

One limitation of our study is that a comparison with non-chemotherapy treated trios of pre- and post-cystectomy tumors with matched normal tissue was not available, but may shed further insight on chemotherapy-specific effects. Overall, how-ever, expanded analysis of pre-post chemotherapy matched pri-mary site tumor samples in MIBC contextualized with phenotypic response data will inform new investigations into exposure-related mutagenesis, treatment strategies, and patient risk-stratification modalities in this disease.

## Methods

**Patient population and samples**. Eligible patients were diagnosed with muscle-invasive bladder cancer and treated with neoadjuvant cisplatin-based

chemotherapy (NAC) followed by cystectomy. A total of 101 patients from 2 clinical trials of NAC[8,9] and a combined retrospective cohort[7] from 3 different academic medical centers (Memorial Sloan Kettering Cancer Center, Fox Chase Cancer Center, Dana Farber Cancer Institute) were analyzed, and details of these populations and studies have been described previously[7–9]. Informed consent was obtained from all patients. In total 56 out of 101 patients were non-responders to NAC with pT1 + disease left at cystectomy, and 46 of those patients had sufficient tumor tissue in study specimens of formalin-fixed, paraffin-embedded (FFPE) tissue sections from cystectomy samples that were subsequently sequenced. In our final cohort, all patients had pT2 + (muscle-invasive) disease at cystectomy. Mat-ched germline DNA was extracted from either peripheral blood mononuclear cells or histologically normal nonurothelial tissue.

**DNA extraction and exome sequencing**. DNA extraction, whole exome library prep and sequencing was performed for samples from DFCI and MSK ($n = 18$ matched pre-post tumors), as previously described[7]. Slides were cut from FFPE blocks and examined by a board-certified pathologist to select high-density cancer foci and ensure high purity of cancer DNA. Biopsy cores were taken from the corresponding tissue block for DNA extraction. DNA was extracted using Qiagen's QIAamp DNA FFPE Tissue Kit Quantitation Reagent (Invitrogen). DNA was stored at −20 °C. Whole exome capture libraries were constructed from 100 ng of DNA from tumor and normal tissue after sample shearing, end repair, and phosphorylation and ligation to barcoded sequencing adaptors. Ligated DNA was size selected for lengths between 200 and 350 bp and subjected to exonic hybrid capture using SureSelect v2 Exome bait (Agilent). The sample was multiplexed and sequenced using Illumina HiSeq technology. Samples from FCCC ($n = 12$ matched pre-post tumors) were sequenced using Illumina library preps. The Illumina exome specifically targets ~ 37.7 Mb of mainly exonic territory made up of all targets from our Agilent exome design (Agilent SureSelect All Exon V2), all coding regions of Gencode V11 genes, and all coding regions of RefSeq gene and KnownGene tracks from the UCSC genome browser (http://genome.ucsc.edu). The Illumina exome uses Illumina's in-solution DNA probe based hybrid selection method that uses similar principles as the Broad Institute-Agilent Technologies developed in-solution RNA probe based hybrid selection method[47,48] to generate Illumina exome sequencing libraries. Pooled libraries were normalized to 2 nM and dena-tured using 0.2 N NaOH prior to sequencing. Flowcell cluster amplification and sequencing were performed according to the manufacturer's protocols using either the HiSeq 2000 v3 or HiSeq 2500. Each run was a 76 bp paired-end with a dual eight-base index barcode read. Data were analyzed using the Broad Picard Pipeline which includes de-multiplexing and data aggregation.

**Quality control and variant calling**. Initial exome sequence data processing and analysis were performed using pipelines at the Broad Institute. After alignment from the Broad Picard Pipeline, BAM files were uploaded into the Firehose infrastructure[49] which managed intermediate analysis files executed by analysis pipelines. Sequencing data were incorporated into quality-control modules in Firehose[49] to compare the tumor and normal genotypes and ensure concordance between samples. Out of samples from 46 patients, 9 were abandoned due to low input material, 1 was black-listed due to non-matching fingerprinting, 2 excluded due to high estimates of tumor contamination[50], 2 had inadequate coverage for high-confidence mutation calling ( < 40x tumor average coverage), and 2 were removed due to low tumor purity ( < 10% tumor cells and no matched mutations in significantly mutated genes[23]), yielding 30 total pairs of pre and post treatment tumors for analysis.

The MuTect algorithm[51] was applied to identify somatic single-nucleotide variants in targeted exons. Strelka[52] was applied to identify small insertions or deletions. Alterations were annotated using Oncotator[53].

**Neoantigen prediction**. HLA-type was inferred using POLYSOLVER[54], which uses a normal tissue bam file as input and employs a Bayesian classifier to deter-mine genotype for each patient. Neoantigens were predicted for each patient by defining all novel amino acid 9mers and 10mers resulting from mutations (after filtering out mutations with < 3 supportive reads or < 30 total reads at the position) and determining whether the predicted binding affinity to the patient's germline HLA alleles was < 500 nM using NetMHCpan (v2.4)[55].

**Purity and ploidy, clonal and subclonal mutational calls**. Purity and ploidy was estimated using the ABSOLUTE algorithm[56], which integrates variant allele fre-quency distributions and copy number variants to estimate absolute tumor purity and ploidy and infer cancer cell fraction (CCF), the proportion of cancer cells in the sample which contain each mutation. An extension of ABSOLUTE[56] was used to infer a phylogenetic tree with clones, subclones, and evolutionary relationships in pre and post treatment tumor samples, as described in detail in Brastianos et al.[29] Briefly, clones and subclones were determined through Markov Chain Montecarlo (MCMC) sampling using Dirichlet process Mixture Models on pre- and post-treatment mutation CCFs, an approach which assigns mutations to subclones without pre-specifying the number of subclones. Mutations inferred to be in a subclone with CCF = 1.0 were called "clonal" while those inferred to be in a subclone with CCF < 1.0 were called "subclonal". For each subclone, two CCFs

were inferred: the CCF in the pre-treatment tumor and CCF in the post-treatment tumor.

**Changes in mutational and neoantigen load.** Differences in tumor purity and depth of coverage can confound analyses of differences in mutations between two tumor samples. Therefore in our comparison between pre- and post-treatment mutations we only considered those mutations which either were detected or had power > 80% to be detected in both tumors (Supplementary Fig. 16). Changes in mutational load and neoantigen load were calculated using a paired t-test of changes in paired samples with a null hypothesis of a difference of 0. Pre-treatment mutation load vs. shared mutation load, and shared mutation load vs. post-treatment mutation load were compared using Mann–Whitney–Wilcoxon tests. A one-way ANOVA was used to compare the proportion of lost mutations among all, clonal, and sub-clonal mutations. $p < 0.05$ was considered to be statistically significant.

**Heterogeneity definition.** We defined and assessed heterogeneity in two ways: (1) proportion of mutations that were inferred as subclonal (as defined above), calculated separately for pre- and post- treatment tumors, and (2) the total number of subclones inferred by the MCMC process described above, which includes subclones private to pre- or post-treatment tumors (where CCF in one tumor = 0) and subclones shared between tumors (CCF in both tumors > 0).

**Mutational signature analysis.** We developed two separate algorithms for mutational signature analysis, one for discovery of mutations, and the second for assessing mutational signature activity of individual tumor samples given a set of mutational signatures.

**Discovery of new mutational signatures.** Following Alexandrov et al.[13], we represented each SNV as a single nucleotide variant (e.g., C > A, C > T, by convention beginning with the pyrimidine) within a trinucleotide context (e.g., "GCT > GAT"), yielding 96 different possible motifs. Each tumor was then represented as a distribution of mutations within these motifs; for the cohort generating a matrix with the 96 trinucleotide contexts in rows and each tumor represented in a column. We then performed a non-negative matrix factorization[57] (NMF) to generate mutational signatures and inferred activity of these mutational signatures in each tumor. The optimal rank (number of mutational signatures) was inferred after manually examining cophenetic coefficients[58], residuals, and residual sum of squares for 50 NMF runs at ranks 2–8, as well as comparing discovered signatures to previously discovered signatures using a cosine similarity measure. High cophenetic coefficients, low residuals, low residual sum of squares, and high cosine similarity to previous signatures were preferred. We used the R-packages SomaticSignatures[59] and NMF[60] with the Brunet update method[58]. Since NMF is non-deterministic, we performed 200 independent NMF runs for a given rank and chose the resulting mutational signatures and signature activity per tumor from the NMF run with the minimum residual error.

We compared our discovered signatures to the 30 existing discovered and validated signatures in COSMIC[16]. Cosine similarity was used to compare our discovered signatures and previously detected signatures, using a threshold of 0.85. We also manually visualized and inspected similarities in mutational motifs between our signatures and COSMIC signatures. We manually examined all generated mutational signatures from the 200 runs to evaluate run variance in mutational signature similarities to COSMIC mutational signatures.

**Inference of mutational signature activity in individual tumors.** Given a set of mutational signatures, inferring the activity of each mutational signature within individual tumors was performed by modifying the NMF multiplicative update process[15,58]. We randomly initialize a starting activity matrix $H_0$ and update $H_i$ to $H_{i+1}$ via the multiplicative update rule given for $H$[15]. W remains fixed and so is not updated. We keep track of the Frobenius norm of the error given by V-WH$_i$ at each iteration. We terminated the update process when error vs. iterations demonstrated horizontal asymptomatic behavior, which we defined as when 1/20 of the difference of the error between V-WH$_{i-20}$ and V-WH$_i$ is below a given threshold (0.0001). The resulting matrix $H_i$ is a good representation of the activity of the signatures in W.

**Discovery and validation of cisplatin mutational signature.** To generate candidate mutational signatures for a cisplatin-induced mutational signature, we used our discovery process independently on 30 matched pre-treatment tumors and the 30 matched post-treatment tumors, and discovered an additional mutational signature in the post-treatment tumors which did not match previously discovered signatures in COSMIC. For comparison, we used a signature of cisplatin-induced mutational activity from whole genome sequencing of a chicken lymphoblast (DT40) line[18], which we normalized to a human exome context by adjusting for the difference in tri-nucleotide context frequencies in chicken and human genomes[18], and between human genomes and exomes using genome and coding intervals from hg19. To compare similarities in mutational activity between the DT40 signature and our candidate cisplatin-induced mutational signature, we

replaced our candidate signature with the DT40 signature and inferred mutational signature activity in the post-treatment tumors (Supplementary Fig. 5). We then calculated a Pearson correlation coefficient between the inferred activities of our candidate signature and the DT40 signature.

To evaluate the significance of this correlation in activity, we repeated the process of replacing our candidate cisplatin mutational signature with another signature, inferring mutational activity of that signature, and correlating the inferred activity between the two. We generated random signatures in two ways: (1) randomly permuting the 96 trinucleotide motifs in the DT40 signature, and (2) taking random linear combinations of two randomly chosen COSMIC signatures. This corresponds to a null hypothesis that our candidate signature is a linear combination of pre-existing, known mutational signatures. Repeating this process generates a "null" distribution of Pearson correlation coefficients, and we calculated an empiric "p-value" that is simply the proportion of simulated correlation coefficients better than our observed correlation coefficient (Supplementary Fig. 6a, b).

To evaluate a transcriptional strand bias, we represented each mutation as one of 192 potential motifs rather than 96 (e.g., treating C > T and G > A as separate motifs), ran the mutational signature discovery process on this matrix (Supplementary Fig. 7), and looked for asymmetry in the complementary mutational signatures motifs (e.g., imbalance between C > T and G > A motifs within the corresponding trinucleotide context). Without a transcriptional strand bias, we would expect similar distributions, e.g., CCC > CTC would have the same distribution as GGG > GAG, and calculated a p-value assuming a binomial distribution with probability = 0.5 for each SNV context (C > A, C > T, C > G, T > A, T > C, T > G).

To evaluate mutational signatures active during chemotherapy, we generated per patient a set of "new" mutations that were detected only in the post-treatment tumor sample, removing those mutations that had been also detected in the pre-treatment tumor. These mutations represented both mutations from intratumoral heterogeneity and sampling differences, as well as mutations induced by cisplatin-based chemotherapy. We performed our mutational discovery procedure on this data set and compared the discovered signatures to our candidate mutational signature.

To evaluate the association of a mutational signature with subclonality, we first inferred each mutation as subclonal or clonal, as above. For each mutation within a tumor, credit was assigned to signatures weighted by (1) the specific signature activity for the trinucleotide context of the mutation and (2) the relative inferred activity level of each signature in the tumor. For example, for a clonal CCT- > CTT mutation in a tumor with inferred activity levels [10, 90] of two mutational signatures S1 and S2 with CCT- > CTT activity of [0.05] and [0.01] respectively, S1 would receive a clonal mutation attribution of $c = 10{\times}0.05/(10{\times}0.05 + 90{\times}0.01)$, and S2 would receive (1-$c$). Iterating this process for every mutation in the cohort resulted in clonal and subclonal mutational activity for each signature. We then assessed the statistical significance of differences in proportion of subclonal mutations per signature using a chi-squared test with DF = (number of signatures − 1).

To generate the cisplatin-induced mutational signature in C. elegans[38] in a human context, we generated the overall frequency of all 96 trinucleotide contexts in the C. elegans genome (ce10), then normalized the frequency of observed mutations in C. elegans exposed to cisplatin by the ratio of each trinucleotide context frequency in C. elegans to the corresponding frequency in the human exome (Supplementary Fig. 15). We then compared this mutational signature with the cisplatin-induced mutational signature in DT40 cells and our candidate cisplatin-induced mutational signature in MIBC (Supplementary Fig. 15).

To generate mutational signatures in the Faltas[6] cohort of 16 patients with pre-post chemotherapy matched tumors, we followed the same procedure for de-novo signature discovery as previously described, using only those mutations found exclusively in the matched post-treatment tumors (enriched for chemotherapy-induced mutations) for analysis. Detailed mutation data from 15 patients was available. During QC, we excluded one post-chemotherapy tumor sample from the lung (WCM077_3) which did not match the pre-chemotherapy primary bladder tumor (WCM077_1) (99.6% non-matching mutations; common mutations only in non-driver genes TCHH and TNNI).

**Discovery of resistance biomarkers.** Mutational Significance: We used MutSig2CV[23] to identify significantly mutated genes across the cohort of resistant tumors. For each altered gene in the pre-treatment tumors, we calculated a p-value of mutational significance, then calculated the percentage of such mutations identified in the pre-treatment tumor that continued to be detected in the post-treatment sample. We further calculated a p-value of mutational significance considering only those mutations private to the post-treatment tumor. Adjustment for hypothesis testing was performed using a Benjamini–Hochberg FDR of 0.1.

Copy Number Alterations and Evolution: Total copy number alterations for individual tumors were inferred using adaptations of a binary segmentation algorithm[61,62] (CapSeg) comparing fractional exon coverage for tumor segments to a panel of normal samples, generating exomic segments and segment copy number. Copy number data were inspected visually and manually for focal amplifications and deletions, and genes were annotated with Oncotator[53]. For allelic copy numbers, heterozygous SNPs were identified and integrated with the binary

segmentation algorithm (Allelic CapSeg), and further adjusted for tumor purity and ploidy[56]. We then called allelic amplifications and deletions, following previously described criteria[29] integrating segment focality and the revised allelic copy number.

**Survival analysis**. To analyze the association between intratumoral heterogeneity (defined above) and overall survival, we performed a survival analysis using both a Cox Proportional Hazards model and a log-rank test dividing the cohort into "high" and "low" heterogeneity samples. For the log-rank test, the cohort was divided in half using a threshold of 0.2 proportion of mutations inferred to be subclonal (as defined above) in the pre- and post-treatment setting (Supplementary Fig. 12a, b), and a threshold of 6 subclones (Supplementary Fig. 12c). To adjust for clinical covariates, we performed a multivariate analysis adding age, gender, pre-treatment pathologic staging (T and N staging), and type of chemotherapy (GC or MVAC) to the Cox PH model (Supplementary Table 1a–c).

**Data availability**. All BAMs for 30 matched pre and post-treatment tumors will be deposited in dbGAP (phs000771.v2.p1). Code to regenerate figures is available on request.

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

## Acknowledgements

This work was funded by the Starr Cancer Consortium (J.R., E.V.A.), NIH 3P30CA006927-50SD (E.R.P.), NIH/NCI P30 CA008748 (Memorial Sloan Kettering Cancer Center Support Grant), NIH/NCI P30 CEA006927 (Fox Chase Cancer Center Support Grant), K08 CA188615 (E.V.A.), and the Damon Runyon Foundation (E.V.A.).

## Author contributions

D.L., P.A., D.K., K.M., A.W.-T., S.M., M.Y.T., J.K., G.I., G.G., J.H.-C., L.A.G., S.L.C., J.B., E.R.P., J.E.R., E.V.A. designed the study. D.L., D.K., B.R., D.M., B.R., G.H., J.K., E.V.A. carried out the analysis. S.M., C.C., G.I., H.A.-A., E.D., D.Y.T.C., R.K.A., J.H.-C., J.B., E.R.P., J.E.R., E.V.A. procured samples and generated data. D.L., P.A., J.B., E.R.P., J.E.R., E.V.A. wrote the manuscript. All authors read and approved the final version of the manuscript.

## Additional information

**Competing interests:** The authors declare no competing financial interests.

