## [Peer Review File · Nature Communications]

Reviewers' comments:

Reviewer #1 Expert in bladder cancer genetics and bioinformatics:

The investigators studied the effects of chemotherapy in a set of matched pre-chemotherapy and post-chemotherapy primary tumors. This approach allowed for direct comparisons of the mutational characteristics of chemotherapy-resistant bladder cancer. Overall, the study is well done but the conclusions are generally not supported by the data and the findings are not novel.

Major points

- **Methods.** The methods section indicates that the authors sequenced both Frozen and paraffin-embedded samples. Were there any differences in the sequencing metrics between frozen and FFPE samples? Was there an over-representation of either frozen or FFPE samples in either the pre-chemotherapy or post-chemotherapy groups and if so, can the authors show that these changes will not affect the results?
- The authors identified a specific mutational signature, which is ascribed to cisplatin activity. While the results are certainly interesting they raise some questions that the authors should address.

1-The patients included in the study were treated with several chemotherapy regimens including MVAC and GC. Each additional drug has a distinct DNA-damaging mechanism distinct from the cisplatin-induced intra- and inter-strand DNA crosslinking. The mutational effect of these DNA damaging agents potentially contributed to the observed mutational signature. While Szikriszt et al's paper cited by the authors of this study shows that Gemcitabine and Doxorubicin have no measurable mutagenic effect; it did not address the effect of Vinblastine or Methotrexate. It is also worth noting that the mutational signature in the Szikriszt DT40 lymphoblast cell line model tested the effects of each drug as monotherapy whereas these patients were clearly treated with combination-chemotherapy. The mutational effects of such combinations are potentially different from the effects of these drugs when used as single agents.

2- The authors cite the mutational data showing cisplatin's effect in *C. Elegans* but do not show a bioinformatic comparison of mutational patterns of cisplatin-induced DNA damage between the *C. Elegans* and patient data.

3- Several studies including Findlay et al.

<https://www.nature.com/articles/ncomms11111> , Murugaesu et al.

<http://cancerdiscovery.aacrjournals.org/content/5/8/821>)

had already identified an increase in C>A mutations secondary to cisplatin-based chemotherapy-treated tumors. This study adds more granularity to the features of this signature (e.g., the detection of transcriptional strand bias), however, the basic observation is not novel.

- The correlation between the numbers of subclones in the tumor predicts worse survival in MIBC patients. As a prognostic indicator, was this adjusted for other clinical factors such as stage, response to chemotherapy, age, and performance status in multivariate analysis? It would also be beneficial to clearly explain how the number of clones was calculated and what thresholds were used to define a "clone".

- The finding of focal PDL1/2 amplification in one patient's tumors post-chemotherapy is interesting. If the author's intention is to describe a focal amplification of two neighboring genes PDL1 and PDL2, it would be helpful to spell that out initially. Were there any other genes included in this focal deletion? The p24 locus is very close to the 9p21 locus which harbors three important tumor-suppressor genes and also to the type I-IFN locus. Were any of the genes in these loci also affected? If so this could alter the interpretation of the selection of PDL1/2 amplification under treatment. Was there any evidence that this focal PDL1/2 resulted in more immunosuppression and a decrease in Tumor-infiltrated lymphocytes or CD8+ mRNA signatures in this tumor?

Minor points

- "We performed whole exome sequencing on matched pre-chemotherapy biopsy tissue, post-chemotherapy cystectomy tumor tissue, and peripheral blood". Suggest clarifying that peripheral blood (peripheral mono-nuclear blood cells) were used as a germline comparison. This is mentioned in the methods section but deserves mention in the body of the manuscript as well.

Reviewer #2 Expert in bladder cancer genetics:

Perioperative cisplatin-based combination chemotherapy remains the standard of care for patients with muscle-invasive bladder cancer. Recent work has established that immune checkpoint blockade with blocking anti-PD1 or –PDL1 antibodies is active in a significant fraction of patients with cisplatin-refractory disease. Because cisplatin is a DNA damaging agent, there is strong interest in determining how cisplatin might affect the subsequent (or coincident) response to immunotherapy by altering the spectrum of tumor DNA mutations that are recognized by the T cells that are reactivated by these agents. Here the investigators characterized the effects of cisplatin-based combination chemotherapy on tumor mutational profiles by performing whole exome sequencing on a relatively large set of tumors (n = 30) collected within the context of two completed clinical trials augmented by a third group of tumors collected from patients off protocol. They report two novel findings – that cisplatin exposure is associated with the emergence of a novel mutational signature that the authors attribute to cisplatin-induced DNA damage and that post-treatment mutational heterogeneity is associated with worse clinical outcomes. They also show that chemotherapy caused no significant changes in overall mutational load (a surrogate for neoantigen burden) or tumor clonal fraction, which may indicate that chemotherapy has no strong, DNA-based effects on tumor immune recognition. The study is extremely timely and has high potential impact.

However, one overall concern is that this paper follows another high-profile study that focused on the same subject (Faltas et al, Nat Genet 48: 1490, 2016), and it is not clear that key conclusions of the previous study are supported by the data presented here. For example, although both papers concluded that the mutational profiles of matched tumors were (surprisingly) heterogeneous, the Faltas paper argued that copy number profiles were more similar, and the papers reached different conclusions regarding the effects of chemotherapy on clonality (the Faltas paper concluded that it increased). In addition, although both papers found evidence of cisplatin exposure in the post-treatment mutational profiles, the Faltas paper did not present evidence for a new cisplatin-associated mutational signature but rather proposed that chemotherapy increased APOBEC-mediated mutagenesis. Unfortunately, neither study included/includes a control cohort of matched tumors collected at TURBT and cystectomy from patients who received no neoadjuvant chemotherapy, which necessitates that the effects of chemotherapy on tumor mutational burden and clonality must be inferred.

Therefore, while there is very high enthusiasm for this manuscript, these issues, and the following points, should be addressed.

1. One of the other major conclusions in the Faltas study was that tumor clonal branching (and therefore the emergence of heterogeneity) occurs very early in the process of tumor initiation and progression. They included figures that contained the results of tumor phylogenetic analyses, and

these figures, which have become fairly standard in studies like these, were very helpful in visually documenting tumor heterogeneity in their cohorts. The authors of this study should include similar descriptions of clonal divergence in this study (presumably in the Supplement).

2. Similarly, the copy number analyses performed in the previous study provided important information that informed conclusions about tumor heterogeneity. The authors should weigh in on the subject of whether CNAs were similar in the matched pre- and post-treatment tumors.

3. The analyses of the DT40 cisplatin signature are creative and definitely support the authors' conclusions about the presence of a specific cisplatin-associated mutational signature in tumors exposed to chemotherapy. However, as they themselves point out, generating their own signature(s) from human bladder cancer cells exposed to cisplatin in vitro or in vivo and showing that this signature was similar to the one observed in patients would have been even more attractive. Although it would take time, it doesn't seem that this kind of experiment would be very difficult to do. In addition, if possible, the authors should look to see whether their signature is also present in the Faltas dataset of 16 matched tumors.

4. The observation that post-treatment tumor heterogeneity is associated with poor outcomes is extremely interesting. Did any of these patients go on to receive immune checkpoint blockade? Are the effects on survival potentially confounded by this variable?

5. Pretreatment tumor heterogeneity was also almost associated with poor outcomes ($p = 0.052$). Likewise, there also could be a correlation between pre- and post-treatment heterogeneity that is not statistically significant here because the sample size is small. These points should be acknowledged.

6. Some anecdotal associations are made between particular post-treatment mutational patterns and resistance. However, it is not clear that there is direct evidence supporting the roles of E2F3, JUN, FBXW7, and PDL1 in chemoresistance, and the high levels of heterogeneity post therapy argue against a strong selective pressure. Absent more direct functional evidence, these correlations do not seem to add much value to the study.

7. The effects of chemotherapy on predicted neoantigen burden (and if possible, TCR clonal diversity) should be discussed. Descriptions of overall tumor mutational burden do not really directly address the topic that really drives interest.

8. Is there any evidence that chemotherapy selected against DDR mutations?

9. Several of the references in the reference list require updating or reformatting.

Reviewer #3 Expert in genomic heterogeneity and clonal evolution:

This study applied exome sequencing to 30 pairs of pre-treatment bladder cancer biopsies and post-chemotherapy samples to investigate the impact of chemotherapy on genomic landscapes and mutation signatures. The authors show that mutation loads do not increase after chemotherapy but they identified a novel mutational signature which they attribute to cisplatin induced DNA damage. Overall, this is a very interesting study and the conclusions appear to be well justified. However, a few points should be improved before considering it for publication in Nature Communications.

Major criticism:

The authors only compare their new signature to that published in DT40 cells but do not make use of existing data describing a cisplatin mutation signature in *C. elegans* which they reference (36) – the analysis should be further extended to compare their results with what is described in this publication.

It remains unclear why the mutation load is frequently lower in samples taken after chemotherapy than at initial biopsy. This is particularly surprising as new mutations are induced through chemotherapy. Could this be a result of subclones which were depleted by the chemotherapy?

There is no diagram/figure of the driver mutations or driver genes which are mutated or amplified/homozygously deleted in these tumours. This would be useful for the purpose of comparison of the results to other studies.

Minor criticism:

The authors state that all tumours were non-responders. How was this assessed? And did some tumours progress during therapy?

Reviewers' comments:

Reviewer #1 Expert in bladder cancer genetics and bioinformatics:

The investigators studied the effects of chemotherapy in a set of matched pre-chemotherapy and post-chemotherapy primary tumors. This approach allowed for direct comparisons of the mutational characteristics of chemotherapy-resistant bladder cancer. Overall, the study is well done but the conclusions are generally not supported by the data and the findings are not novel.

We appreciate the reviewer's comments here and below, and we feel that by addressing them in this revised manuscript, it has strengthened the underlying methodology and highlighted the novel findings, as outlined below.

Major points

- **Methods. The methods section indicates that the authors sequenced both Frozen and paraffin-embedded samples. Were there any differences in the sequencing metrics between frozen and FFPE samples? Was there an over-representation of either frozen or FFPE samples in either the pre-chemotherapy or post-chemotherapy groups and if so, can the authors show that these changes will not affect the results?**

Thank you for the question. Upon clarification among the 3 centers that provided samples (DFCI/MSKCC/FCCC), no samples included in this cohort were fresh frozen; all were FFPE samples. We have now **amended our text** (page 13, paragraph 2) to correctly reflect this. As the reviewer suggests, formalin fixation induces C>T artifacts through cytosine deamination; we leveraged a filter (<http://genomics.broadinstitute.org/data-sheets/error-proofing-exome-sequencing-somatic-variant.pdf>) based on a read orientation bias to correct for this concern in this cohort.

- **The authors identified a specific mutational signature, which is ascribed to cisplatin activity. While the results are certainly interesting they raise some questions that the authors should address.**

1-The patients included in the study were treated with several chemotherapy regimens including MVAC and GC. Each additional drug has a distinct DNA-damaging mechanism distinct from the cisplatin-induced intra- and inter-strand DNA crosslinking. The mutational effect of these DNA damaging agents potentially contributed to the observed mutational signature. While Szikriszt et al's paper cited by the authors of this study shows that Gemcitabine and Doxorubicin have no measurable mutagenic effect; it did not address the effect of Vinblastine or Methotrexate. It is also worth noting that the mutational signature in the Szikriszt DT40 lymphoblast cell line model tested the effects of each drug as monotherapy whereas these patients were clearly treated with combination-chemotherapy. The mutational effects of such combinations are potentially different from the effects of these drugs when used as single agents.

The reviewer raises important points that (1) we do not know the effects of combination chemotherapies, and (2) there were two different chemotherapy regimens that patients were treated with. In our cohort, only 8 patients were treated with some form of MVAC combination therapy, which we found insufficient data to infer an independent mutational signature in these patients. To distinguish mutational signatures generated by these two chemotherapeutic regimens, future studies would ideally take two different approaches: (1) sequencing of a larger number of matched pre- and post-MVAC and GC treated tumors (e.g. SWOG 1314, a clinical trial comparing GC vs. dose-dense MVAC in the neoadjuvant setting); (2) functional experiments with combination vs. single-agent therapies in pre-clinical human models. We have acknowledged this excellent point in the **revised Discussion (starting page 11, paragraph 3)**.

2- The authors cite the mutational data showing cisplatin's effect in C. Elegans but do not show a bioinformatic comparison of mutational patterns of cisplatin-induced DNA damage between the C. Elegans and patient data.

Thank you for the suggestion. We performed new analyses and generated the mutational spectrum of C. Elegans based on 27 worms exposed to cisplatin at the highest dose in the Meier et al paper and adjusted for the relative trinucleotide frequencies in C. elegans vs. human exome (new Supplemental Fig 15). This demonstrated that C->A mutations in the C.C and C.T context were the most abundantly observed mutations, but this signature was not similar to either our candidate cisplatin signature (cos sim = 0.36), or the DT40 cisplatin signature (cos sim = 0.44). This highlights the challenges of using these non-human pre-clinical models for comparison; mutational signatures for DNA-damaging agents may differ significantly across species. We chose to compare the chicken lymphoblast signature to our human samples because the chicken is closer evolutionarily and genomically compared to the invertebrate worm C. elegans, and have **revised our text (page 11, paragraph 2)** to include this finding.

3- Several studies including Findlay et al.

<https://www.nature.com/articles/ncomms11111> , Murugaesu et al. <http://cancerdiscovery.aacrjournals.org/content/5/8/821>)

had already identified an increase in C>A mutations secondary to cisplatin-based chemotherapy-treated tumors. This study adds more granularity to the features of this signature (e.g., the detection of transcriptional strand bias), however, the basic observation is not novel.

In both of these studies, an increase in proportion of C>A mutations were observed in post-chemotherapy samples (n=5 in the Murugaesu et al study, and n = 30 in the Findlay et al study). In the Murugaesu study, patients with EAC were treated with 1-4 cycles of neoadjuvant ECX (epirubicin, cisplatin, capecitabine) or EOX (epirubicin, oxaliplatin, capecitabine) as compared to their matched pre-chemotherapy samples. In the Findlay et al study, patients were treated with two doses of neoadjuvant 5-FU and oxaliplatin. Thus, the clinical context, disease, and accompanying agents used were different than in this study, and the effect of these differences on the mutational signature generated is unknown, as previously discussed. Further, oxaliplatin, in particular, has recently been shown to have a mechanism of action other than DNA-damage for its anti-tumor effects (Bruno Nature Medicine 2017).

In our inferred cisplatin-based mutational signature, we infer additional features, including (1) transcriptional strand bias; (2) association with subclonal mutations, (3) additional C>T mutations primarily in a CpC context that are not consistent with other existing mutational signatures (aging, APOBEC), and (4) T>A mutations primarily in a CpT context. We believe that these additional characteristics provide evidence towards the validity of this inferred signature as a cisplatin-associated mutational signature, as well as provide specificity of exposure to cisplatin. We appreciate the comments and **revised our Discussion (page 11, paragraph 2)** to acknowledge this distinction.

• The correlation between the numbers of subclones in the tumor predicts worse survival in MIBC patients. As a prognostic indicator, was this adjusted for other clinical factors such as stage, response to chemotherapy, age, and performance status in multivariate analysis?

We appreciate the excellent suggestion and have performed multivariate analysis in this revised manuscript. All patients were PS 0 or 1 (as candidates for chemotherapy). After adjusting for age, gender, post-chemotherapy pathologic T and N staging from cystectomy (all patients were M0 or Mx), and type of chemotherapy (gemcitabine + cisplatin vs. MVAC), we found that pre-treatment heterogeneity was not statistically significantly associated with overall survival, but post-treatment heterogeneity and the number of subclones, and integration of pre- and post-treatment heterogeneity, continued to be associated with worse overall survival. We have included this information in **revised Supplemental Table 3** and **revised the Methods (page 21, paragraph 3)** and **Results (page 9, paragraph 1)** accordingly.

It would also be beneficial to clearly explain how the number of clones was calculated and what thresholds were used to define a “clone”.

We apologize for the lack of clarity. Clones were defined using a Dirichlet clustering process on the inferred cancer cell fraction (CCF) of mutations in pre- and post- treatment tumor samples, as described in detail by Brastianos and colleagues (Brastianos Cancer Discovery 2015), an approach that assigns mutations to clusters without pre-defining the number of clusters. The total number of clusters thus inferred was interpreted as the number of clones. We have **revised the Methods (page 15, paragraph 3)** to better explain our approach.

• The finding of focal PDL1/2 amplification in one patient’s tumors post-chemotherapy is interesting. If the author’s intention is to describe a focal amplification of two neighboring genes PDL1 and PDL2, it would be helpful to spell that out initially. Were there any other genes included in this focal deletion? The p24 locus is very close to the 9p21 locus which harbors three important tumor-suppressor genes and also to the type I-IFN locus. Were any of the genes in these loci also affected? If so this could alter the interpretation of the selection of PDL1/2 amplification under treatment. Was there any evidence that this focal PDL1/2 resulted in more immunosuppression and a decrease in Tumor-infiltrated lymphocytes or CD8+ mRNA signatures in this tumor?

Thank you for the excellent suggestions. CDKN2A/B/D located at the 9p21 locus were not involved in the amplified segment containing PD-L1/2 and JAK2, nor were any of the type I-IFN gene locus.

Regarding immune response in the PDL1/2 amplified tumor, we unfortunately do not have histologic data, but microarray data was available for a subset of pre- and post-treatment FCCC samples (Choi Cancer Cell 2014), including the pre-treatment PDL1/2 amplified tumor. We performed CIBERSORT (Newman Nature Methods 2014), an algorithm to deconvolute immune cell contents from RNA sequencing data, and found that this pre-treatment sample did have high relative inferred levels of M2 macrophages (associated with immunosuppression, **Response Fig 1**). However, microarray data for the matched post-treatment sample was not available, so the effect of chemotherapy on the immunological profile of this tumor could not be analyzed, though we agree that this type of investigation may shed light on potential interactions and synergies between chemo- and immune- therapies. Ultimately, studies incorporating clinical response to immunotherapy with integrated histologic, transcriptome, and genomic data would best elucidate these relationships. However, PD-L1 amplification is a defining feature of classical Hodgkin's lymphoma (Roemer JCO 2016), which has a high response rate (e.g. 66% in a heavily pretreated population (Younes Lancet Oncology 2016)) to PD-1 inhibition, suggesting that individuals with these genetic alterations in bladder cancer may be excellent candidates for immunotherapy. We have **revised the Discussion (page 12, paragraph 2)** to further describe these considerations.

Response Figure 1. Inferred immune cell populations in a subset of urothelial carcinoma with microarray data (n = 71 tumors); FCCC-022 (pre-treatment tumor with PD-L1 amplification in the post-treatment setting) is represented as the red dot.

Minor points

- “We performed whole exome sequencing on matched pre-chemotherapy biopsy tissue, post-chemotherapy cystectomy tumor tissue, and peripheral blood”. Suggest clarifying that peripheral blood (peripheral mono-nuclear blood cells) were used as a germline comparison. This is mentioned in the methods section but deserves mention in the body of the manuscript as well.

Thank you for the suggestion. We have **modified the text (page 5, paragraph 1)** to reflect this point.

Reviewer #2 Expert in bladder cancer genetics:

Perioperative cisplatin-based combination chemotherapy remains the standard of care for patients with muscle-invasive bladder cancer. Recent work has established that immune checkpoint blockade with blocking anti-PD1 or –PDL1 antibodies is active in a significant fraction of patients with cisplatin-refractory disease. Because cisplatin is a DNA damaging agent, there is strong interest in determining how cisplatin might affect the subsequent (or coincident) response to immunotherapy by altering the spectrum of tumor DNA mutations that are recognized by the T cells that are reactivated by these agents. Here the investigators characterized the effects of cisplatin-based combination chemotherapy on tumor mutational profiles by performing whole exome sequencing on a relatively large set of tumors (n = 30) collected within the context of two completed clinical trials augmented by a third group of tumors collected from patients off protocol. They report two novel findings – that cisplatin exposure is associated with the emergence of a novel mutational signature that the authors attribute to cisplatin-induced DNA damage and that post-treatment mutational heterogeneity is associated with worse clinical outcomes. They also show that chemotherapy caused no significant changes in overall mutational load (a surrogate for neoantigen burden) or tumor clonal fraction, which may indicate that chemotherapy has no strong, DNA-based effects on tumor immune recognition. The study is extremely timely and has high potential impact.

However, one overall concern is that this paper follows another high-profile study that focused on the same subject (Faltas et al, Nat Genet 48: 1490, 2016), and it is not clear that key conclusions of the previous study are supported by the data presented here. For example, although both papers concluded that the mutational profiles of matched tumors were (surprisingly) heterogeneous, the Faltas paper argued that copy number profiles were more similar, and the papers reached different conclusions regarding the effects of chemotherapy on clonality (the Faltas paper concluded that it increased). In addition, although both papers found evidence of cisplatin exposure in the post-treatment mutational profiles, the Faltas paper did not present evidence for a new cisplatin-associated mutational signature but rather proposed that chemotherapy increased APOBEC-mediated mutagenesis. Unfortunately, neither study included/includes a control cohort of matched tumors collected at TURBT and cystectomy from patients who received no neoadjuvant

chemotherapy, which necessitates that the effects of chemotherapy on tumor mutational burden and clonality must be inferred.

Therefore, while there is very high enthusiasm for this manuscript, these issues, and the following points, should be addressed.

1. One of the other major conclusions in the Faltas study was that tumor clonal branching (and therefore the emergence of heterogeneity) occurs very early in the process of tumor initiation and progression. They included figures that contained the results of tumor phylogenetic analyses, and these figures, which have become fairly standard in studies like these, were very helpful in visually documenting tumor heterogeneity in their cohorts. The authors of this study should include similar descriptions of clonal divergence in this study (presumably in the Supplement).

Thank you for the suggestion. We have included phylogenetic trees of all pre-post tumor pairs in the supplement as **new Supplemental Fig 13**. In all cases, the pre-treatment and post-treatment subclones shared a common ancestor, with additional subclones private to both the pre- and post-treatment setting (e.g. MSK-0745). In some cases (e.g. FCCC-022), there is a subclone seen in both the pre- and post- treatment tumors.

2. Similarly, the copy number analyses performed in the previous study provided important information that informed conclusions about tumor heterogeneity. The authors should weigh in on the subject of whether CNAs were similar in the matched pre- and post-treatment tumors.

We appreciate the suggestion. We have now included a comprehensive visualization of the copy number alterations in pre- and post-treatment samples as **new Supplemental Figure 3**. CNAs were very similar between pre- and post- matched tumors (interquartile range of the difference in % of exome altered is 5.1% (-2.4%, 2.6%)), and we did not detect a statistically significant bias towards increased copy number alterations in pre- or post-treatment samples (mean difference = 0.4%, $p=0.71$, **new Supplemental Fig 4**).

3. The analyses of the DT40 cisplatin signature are creative and definitely support the authors' conclusions about the presence of a specific cisplatin-associated mutational signature in tumors exposed to chemotherapy. However, as they themselves point out, generating their own signature(s) from human bladder cancer cells exposed to cisplatin in vitro or in vivo and showing that this signature was similar to the one observed in patients would have been even more attractive. Although it would take time, it doesn't seem that this kind of experiment would be very difficult to do.

We agree that prospective generation of a cisplatin mutational signature in human bladder cancer cell lines is an important next step, and we are currently planning these experiments. Although conceptually straightforward, significant time is required to optimize experimental conditions (including cell lines, cisplatin concentrations, number of cell divisions, etc) prior to collecting useful data. Given the increased complexity of using human cell systems, in this case specifically in bladder cancer cells, such an experiment would represent a significant advance,

and we are not aware of published reports of prospective generation of NMF-derived mutational signatures in human cell lines. Therefore, while we are actively pursuing these experiments, we believe they are beyond the scope of the work reported here.

In addition, if possible, the authors should look to see whether their signature is also present in the Faltas dataset of 16 matched tumors.

We appreciate this suggestion and in our revised submission, we analyzed the Faltas dataset for existence of a similar cisplatin-induced signature. Out of the 16 patients with matched pre-post treatment samples, data for 15 were available for analyses. We focused our analysis on the mutations found only in matched post-treatment tumors (as we did in our study) to enrich for mutations caused by chemotherapy, and performed a de-novo mutational signature discovery on this data set. We found a similar signature to our candidate cisplatin signature (**new Supplemental Fig 8**, $\text{cos sim} = 0.86$). This validation in the post-cisplatin chemotherapy setting of an independent cohort further suggests that this signature may indeed represent a candidate cisplatin signature in bladder cancer. We have included these findings in the **revised Results (page 8, paragraph 2)**

4. The observation that post-treatment tumor heterogeneity is associated with poor outcomes is extremely interesting. Did any of these patients go on to receive immune checkpoint blockade? Are the effects on survival potentially confounded by this variable?

Thank you for the question. After review of the available clinical data, only 2 patients in our cohort (MSK-0450 and MSK-0754) received immune checkpoint blockade post progression. Neither patient responded to therapy— MSK-0450 had stable disease as the best response and progressed after 8 cycles (24 weeks) of atezolizumab, and MSK-0754 had progressive disease as a best response. Thus, while we cannot definitively conclude that immunotherapy did not affect these two patients' overall survival, it seems unlikely. Finally, when we adjust for post-progression ICB in the multivariate model, our conclusions are unchanged.

5. Pretreatment tumor heterogeneity was also almost associated with poor outcomes ($p = 0.052$). Likewise, there also could be a correlation between pre- and post-treatment heterogeneity that is not statistically significant here because the sample size is small. These points should be acknowledged.

We have now clarified these excellent points in the text. As mentioned above in response to Reviewer 1, after adjusting for additional clinical covariates, pre-treatment heterogeneity is not statistically significantly associated with overall survival, though post-treatment and the number of subclones continue to be associated. These results are included in the revised study.

6. Some anecdotal associations are made between particular post-treatment mutational patterns and resistance. However, it is not clear that there is direct evidence supporting the roles of E2F3, JUN, FBXW7, and PDL1 in chemoresistance, and the high levels of heterogeneity post therapy argue against a strong selective pressure. Absent more direct functional evidence, these correlations do not seem to add much value to the study.

We appreciate the opinion of the reviewer, and agree that additional functional validation as well as analysis of additional clinical cohorts will be important to validate these findings. Given the high response of other tumors with PD-L1/2/JAK2 amplification to immunotherapy (e.g. Younes Lancet Oncology 2016), this finding (albeit in a small subset) in bladder tumors suggests a potential biomarker for response to immunotherapy. E2F3 amplification has been associated with higher grade and stage disease and metastases in bladder cancer (Oeggerli Oncotarget 2004; Bambury BMC Cancer 2015), but we agree has not previously been specifically linked to platinum resistance. FBXW7's role in regulating protein levels of multiple oncogenes (c-MYC, Notch, Cyclin E, and c-JUN among others) suggests a biological mechanism for therapy resistance. Further, alterations in FBXW7 have been associated with worse prognosis and advanced disease in multiple disease settings (reviewed in Cao Medicine 2016) and alterations in FBXW7 have been directly shown to confer resistance to anti-tubulin chemotherapies (Wertz Nature 2011). Our findings suggest that these potential mechanisms and biomarkers should be investigated further in future studies, and we have indicated so in our **revised Discussion (page 12, paragraph 2)**.

7. The effects of chemotherapy on predicted neoantigen burden (and if possible, TCR clonal diversity) should be discussed. Descriptions of overall tumor mutational burden do not really directly address the topic that really drives interest.

Based on this suggestion, we analyzed neoantigens in our cohort and found that the change in neoantigens between matched pre- and post-treatment tumors was highly concordant with the change in mutations (pearson's rho = 0.85, **new Supplemental Fig 2a**). Overall, we found that there was a trend towards decreased neoantigen load (mean change = -41.8 neoantigens, p = 0.07) in post-treatment tumors (**new Supplemental Fig 2b**). Unfortunately, we do not have data on TCR clonal diversity in this population, though we agree that that would be of interest. We have included this finding in our **revised Results (page 5, paragraph 2)**.

8. Is there any evidence that chemotherapy selected against DDR mutations?

As shown in **new Supplemental Fig 9a**, we found no genes that passed statistical significance testing with multiple hypothesis correction that were lost in the post-treatment setting. Based on this suggestion, we further analyzed the incidence of DNA damage repair mutations in pre-treatment tumors compared to post-treatment tumors. We specifically examined damaging alterations in homologous recombination (HR) (**new Supplemental Fig 10**) and nucleotide excision repair (NER) (**new Supplemental Fig 11**) pathway genes, but found similar proportions of pre- and post- treatment tumors with alterations in these pathways (HR: 8/30 tumors with alterations in both pre- and post-treatment settings; NER: 8/30 tumors (pre-treatment) and 7/30 tumors (post-treatment) with alterations). We have included these findings in our **revised Results (page 8, paragraph 3)**.

9. Several of the references in the reference list require updating or reformatting.

We have reviewed these and updated the formats, thank you for the suggestion.

Reviewer #3 Expert in genomic heterogeneity and clonal evolution:

This study applied exome sequencing to 30 pairs of pre-treatment bladder cancer biopsies and post-chemotherapy samples to investigate the impact of chemotherapy on genomic landscapes and mutation signatures. The authors show that mutation loads do not increase after chemotherapy but they identified a novel mutational signature which they attribute to cisplatin induced DNA damage. Overall, this is a very interesting study and the conclusions appear to be well justified. However, a few points should be improved before considering it for publication in Nature Communications.

Major criticism:

The authors only compare their new signature to that published in DT40 cells but do not make use of existing data describing a cisplatin mutation signature in C elegans which they reference (36) – the analysis should be further extended to compare their results with what is described in this publication.

We agree with this suggestion and performed this analysis in the revised draft (see answer to question above).

It remains unclear why the mutation load is frequently lower in samples taken after chemotherapy than at initial biopsy. This is particularly surprising as new mutations are induced through chemotherapy. Could this be a result of subclones which were depleted by the chemotherapy?

We appreciate the question. As shown in Fig 1d and e, most differences in specific mutations between pre- and post-treatment tumors were mutations detected at a subclonal cancer cell fraction, thus differences in mutational load are due to differences in subclonal mutations. As discussed in the **revised text (page 5, paragraph 2)**, this can be the result of depleted subclones + new induced subclones from chemotherapy, or sampling heterogeneity, or both, which we unfortunately cannot distinguish from this data.

There is no diagram/figure of the driver mutations or driver genes which are mutated or amplified/homozygously deleted in these tumours. This would be useful for the purpose of comparison of the results to other studies.

We appreciate the suggestion and have provided this data in **revised Supplementary Fig 1**, and in our **revised Results (page 5, paragraph 1)**.

Minor criticism:

The authors state that all tumours were non-responders. How was this assessed? And did some tumours progress during therapy?

Non-responders were classified based on having persistent disease (pT1+) disease at the time of cystectomy. In fact, all of our cohort had resistant muscle-invasive disease (pT2+) at the time of cystectomy, and we describe this in our **revised Methods (page 15, paragraph 2)**. Prior to

cystectomy, tumor staging is based on a clinical assessment (imaging + bladder biopsy) and it is well known that patients are frequently understaged, making determination of true progression challenging.

Reviewers' comments:

Reviewer #1 (Remarks to the Author):

The authors address all my concerns in their careful rebuttal. I have no further concerns.

Reviewer #2 (Remarks to the Author):

The authors have been highly responsive to the concerns raised in the original reviews, and their revised manuscript is much improved with respect to the original. However, some remaining issues should be addressed.

1. The authors' assertion that this cohort is qualitatively different from the previous cohort analyzed by Faltas et al (Nat Genet) seems a bit disingenuous. The Faltas paper did contain matched primary-primary pre- and post-chemotherapy pairs.
2. The paper still lacks a clear discussion of the conclusions that are similar between the two papers and those that are not.
3. One limitation of both studies is the lack of matched TURBT and cystectomy samples from patients who did not receive NAC. This too should be acknowledged as a (potentially minor) limitation associated with inferring the effects of chemotherapy.
4. The graphs in Figure S3 are visually impactful and helpful. However, it appears that in at least 7/30 cases, the pre and post tumors exhibit significant differences with regard to the specific loci affected by CNVs. The authors should probably comment on this in the manuscript.
5. Although intrinsically interesting, the clinical significance of the cisplatin-associated mutation signatures is unclear. Do the authors have a concrete hypothesis for how they may be clinically significant?
6. What is the significance of the apparent enrichment for TP53, ARID1A, and RB1 mutations in the post-treatment tumors in Figure S9?

Reviewer #3 (Remarks to the Author):

The authors have addressed all of my previous comments.

Reviewers' comments:

Reviewer #1 (Remarks to the Author):

The authors address all my concerns in their careful rebuttal. I have no further concerns.

We thank Reviewer #1 for the helpful comments which have improved the manuscript and analysis.

Reviewer #2 (Remarks to the Author):

The authors have been highly responsive to the concerns raised in the original reviews, and their revised manuscript is much improved with respect to the original. However, some remaining issues should be addressed.

We thank Reviewer #2 for the careful review and helpful suggestions, which we have addressed below and in our manuscript.

1. The authors' assertion that this cohort is qualitatively different from the previous cohort analyzed by Faltas et al (Nat Genet) seems a bit disingenuous. The Faltas paper did contain matched primary-primary pre- and post-chemotherapy pairs.

We believe the primary distinction in these two studies are that the current study focuses exclusively on pre- and immediate post-treatment primary tumors in the context of platinum chemotherapy with increased sample size to characterize this clinical context in more depth, whereas the Faltas et al study performed a broader range of genomic analyses on primary/metastatic and multiregional samples in the setting of chemotherapy but over a longer time period. Both approaches are valid and important, and each yields critical insights (see below). We have modified our text to **clarify this distinction** in our Intro (Page 4, Paragraph 1) and Discussion (Page 11, Paragraph 1)

2. The paper still lacks a clear discussion of the conclusions that are similar between the two papers and those that are not.

We thank the reviewer for the suggestion. Regarding primary pre- and post- treatment tumors, the Faltas et al study finds substantial intra-patient mutational heterogeneity, which we also observe in our samples, though there is substantial variance in heterogeneity, with some tumors with a high proportion of shared mutations and very similar CNV profiles, and some with much smaller proportions of shared mutations and different CNV profiles. However, we were able to characterize mutational differences as predominantly subclonal, suggesting that the dominant clones are similar in the majority of cases. Further, the Faltas et al study finds increased clonality of mutations in post-treatment tumors, which we do not observe in our samples. This difference may partially be due to the bottlenecking effect of metastasis in their mixed cohort. We also did not observe the presence of a smoking-associated mutational signature which was identified in the Faltas et al study in our analysis. Finally, due to our larger sample-size focused specifically on the neoadjuvant context, we were able to discover a de-novo mutational signature that may be due to cisplatin-based chemotherapy, which we were able to reproduce in the relevant subset of the Faltas cohort. The primary conclusion from the Faltas study that we could not examine relates to differences between primary and metastatic tumors, as our study was focused on pre/post-treatment bladder-confined tumors exclusively. We have **modified our discussion to reflect these comparisons** (Page 11, paragraph 1 and Page 13, paragraph 2)

3. One limitation of both studies is the lack of matched TURBT and cystectomy samples from patients who did not receive NAC. This too should be acknowledged as a (potentially minor) limitation associated with inferring the effects of chemotherapy.

We agree that this would be a useful comparison group addressing the effect of “natural” evolution (i.e. without chemotherapy) though the time frame between primary TURBT and cystectomy would be much shorter. We have **modified our discussion** (Page 13, paragraph 3) to acknowledge this distinction.

4. The graphs in Figure S3 are visually impactful and helpful. However, it appears that in at least 7/30 cases, the pre and post tumors exhibit significant differences with regard to the specific loci affected by CNVs. The authors should probably comment on this in the manuscript.

We agree that for most tumors, the pre- and post-treatment CNV profile is similar but in some cases there are significant differences. While some of these differences may reflect heterogeneity in sequencing depth and our ability to detect CNVs due to tumor impurity and sequencing differences, some tumors likely reflect biologically significant differences in pre- and post-treatment tumors. We have **modified our text** to comment on this (Page 6, paragraph 2).

5. Although intrinsically interesting, the clinical significance of the cisplatin-associated mutation signatures is unclear. Do the authors have a concrete hypothesis for how they may be clinically significant?

We believe our cisplatin-associated mutational signature is clinically significant in that it suggests that new neoantigens may be generated specifically by cisplatin-based chemotherapy, which may have implications for subsequent immunotherapies or combination regimens. Further, it is notable that there is variance among the post-treatment tumors in terms of number of cisplatin-associated mutations, suggesting that despite all being resistant to NAC, different mechanisms of resistance may be operant in different tumors (e.g. anti-apoptotic adaptations in the high cisplatin-signature tumors vs efflux pumps or enhanced DNA repair mechanisms in the non-cisplatin-signature tumors), which may yield insight into different clinical resistance mechanisms. We have **modified our discussion** (Page 12, paragraph 2) to reflect these considerations.

6. What is the significance of the apparent enrichment for TP53, ARID1A, and RB1 mutations in the post-treatment tumors in Figure S9?

The analysis that Figure S9a represents is most specific for significantly mutated (e.g. driver) genes with alterations that are **lost post-treatment** suggesting susceptibility to therapy (bottom right quadrant of the figure), which we did not observe. *TP53*, *ARID1A*, and *RB1* are significantly mutated genes in the pre-treatment samples that are **not lost** in the post-treatment samples, suggesting association with resistance. However, persistence may also be simply due to the clonality of these mutations in the tumor. We have **modified our text** (Page 8, paragraph 4) to better reflect this analysis.

Reviewer #3 (Remarks to the Author):

The authors have addressed all of my previous comments.

We thank Reviewer #3 for his/her helpful comments.

REVIEWERS' COMMENTS:

Reviewer #2 (Remarks to the Author):

The authors have now thoroughly addressed my previous concerns.